# Differentiating False Positive Lesions from Clinically Significant Cancer and Normal Prostate Tissue Using VERDICT MRI and Other Diffusion Models

**DOI:** 10.3390/diagnostics12071631

**Published:** 2022-07-05

**Authors:** Snigdha Sen, Vanya Valindria, Paddy J. Slator, Hayley Pye, Alistair Grey, Alex Freeman, Caroline Moore, Hayley Whitaker, Shonit Punwani, Saurabh Singh, Eleftheria Panagiotaki

**Affiliations:** 1Centre for Medical Image Computing, Department of Computer Science, University College London, London WC1E 6BT, UK; snigdha.sen.20@ucl.ac.uk (S.S.); v.valindria@ucl.ac.uk (V.V.); p.slator@ucl.ac.uk (P.J.S.); 2Molecular Diagnostics and Therapeutics Group, University College London, London WC1E 6BT, UK; h.pye@ucl.ac.uk (H.P.); hayley.whitaker@ucl.ac.uk (H.W.); 3Department of Urology, University College London Hospitals NHS Foundations Trust, London NW1 2PG, UK; alistair.grey@ucl.ac.uk (A.G.); caroline.moore@ucl.ac.uk (C.M.); 4Department of Pathology, University College London Hospitals NHS Foundations Trust, London NW1 2PG, UK; arya.freeman@ucl.ac.uk; 5Centre for Medical Imaging, University College London, London WC1E 6BT, UK; s.punwani@ucl.ac.uk (S.P.); saurabh.singh@ucl.ac.uk (S.S.)

**Keywords:** prostate cancer, diffusion MRI, false positives, biophysical modeling, deep learning

## Abstract

False positives on multiparametric MRIs (mp-MRIs) result in many unnecessary invasive biopsies in men with clinically insignificant diseases. This study investigated whether quantitative diffusion MRI could differentiate between false positives, true positives and normal tissue non-invasively. Thirty-eight patients underwent mp-MRI and Vascular, Extracellular and Restricted Diffusion for Cytometry in Tumors (VERDICT) MRI, followed by transperineal biopsy. The patients were categorized into two groups following biopsy: (1) significant cancer—true positive, 19 patients; (2) atrophy/inflammation/high-grade prostatic intraepithelial neoplasia (PIN)—false positive, 19 patients. The clinical apparent diffusion coefficient (ADC) values were obtained, and the intravoxel incoherent motion (IVIM), diffusion kurtosis imaging (DKI) and VERDICT models were fitted via deep learning. Significant differences (*p* < 0.05) between true positive and false positive lesions were found in ADC, IVIM perfusion fraction (*f*) and diffusivity (*D*), DKI diffusivity (*D_K_*) (*p* < 0.0001) and kurtosis (*K*) and VERDICT intracellular volume fraction (*f_IC_*), extracellular–extravascular volume fraction (*f_EES_*) and diffusivity (*d_EES_*) values. Significant differences between false positives and normal tissue were found for the VERDICT *f_IC_* (*p* = 0.004) and IVIM *D*. These results demonstrate that model-based diffusion MRI could reduce unnecessary biopsies occurring due to false positive prostate lesions and shows promising sensitivity to benign diseases.

## 1. Introduction

Prostate cancer (PCa) is traditionally diagnosed via digital rectal inspection (DRE) and a prostate-specific antigen (PSA) test, followed by transrectal ultrasound (TRUS)-guided biopsy [1]. Multiparametric MRI (mp-MRI) has recently been introduced as a standard part of the prostate cancer clinical diagnosis pathway [2]—it consists of T1- and T2-weighted images, diffusion-weighted (DW) images and dynamic contrast-enhanced (DCE) imaging. This technique has high sensitivity (90%) but moderate specificity (50%), translating to a high rate of false positive cases [3]. This results in one in two men undergoing mp-MRI having unnecessary uncomfortable biopsies and risking the associated side effects for benign conditions or clinically insignificant cancer [4]. This is a significant issue, as 75% of suspected cancer patients have abnormal mp-MRI findings, and the number of people considered for MRI and biopsy each year is set to increase; therefore, reducing the number of unnecessary negative biopsies is an important clinical problem [5].

Benign pathologies such as atrophy, inflammation and high-grade prostatic intraepithelial neoplasia (PIN) are examples of diseases that cause these false positive results [6]. This is due to the changes these diseases cause to the microstructure, similar to cancer. For example, atrophy is characterized by shrinkage of prostate tissue due to the reduction in cytoplasm prostatic acinar cells and has been associated with prostatic inflammation (swelling of the prostate gland) [7]. High-grade PIN represents the pre-invasive end of the range of cellular proliferations within the lining of prostatic ducts and acini and is considered the most likely precursor of PCa, with most patients developing carcinoma within 10 years [8]. It is critical to discriminate these cases from cancer to avoid unnecessary procedures; however, it is also important to distinguish these diseases from normal tissue and correctly identify the type of the disease [9]. Some of the benign diseases can present with symptoms similar to PCa, such as difficult or frequent urination and pain, requiring treatment of their own [10]. Identifying a unique noninvasive signature for such diseases can lead to early and informed treatments.

DW-MRI is an integral component of mp-MRI due to the unique insight it provides into the tissue microstructure. Changes in histological features, such as the cellular density, size, shape and arrangement, produce contrast in DW-MR images as they all affect tissue-water mobility. Most studies using DW-MRI have focused on calculating the apparent diffusion coefficient (ADC) to distinguish between tumor regions and healthy tissue [11,12]. Typically, ADC values are lower in prostate tumors than in the surrounding tissue, reflecting the highly cellular environment constraining the water mobility. However, the simultaneous dependence of the ADC on a multitude of histological features limits its biological specificity [13], thus reducing its ability to distinguish cancer from similar diseases such as high-grade PIN and hyperplasia, which often appear as false positive cases [14,15]. More sophisticated models have been proposed to improve the sensitivity and specificity of DW-MRI for cancer diagnosis, such as the intravoxel incoherent motion (IVIM) model that separates the pure water diffusion in tissue from the microcirculation of water in capillaries [16]. It has been used to study various cancer types, such as breast [17], prostate [18] and pancreatic [19] tumors, showing improvement in data description over ADC. Another method that has shown greater sensitivity for the discrimination of benign and cancerous prostate tissue in comparison to ADC is diffusion kurtosis imaging (DKI) [20,21]; this technique quantifies the Gaussian and non-Gaussian components of water diffusion in biological tissues [22].

In an attempt to increase biological specificity, multicompartment microstructure models have also been proposed for imaging the prostate. One of the first multicompartment methods for cancer imaging is the Vascular, Extracellular and Restricted Diffusion for Cytometry in Tumors (VERDICT) MRI framework, which is a non-invasive imaging technique for quantifying microstructural features of tumors in vivo. It consists of a specific imaging protocol, as well as a model for the DW-MRI signal [23]. VERDICT allows for estimation of specific tissue properties, such as cell size and packing density. It has been successful in delineating benign from cancerous lesions [24], and preliminary results from the clinical trial INNOVATE [25] reveal that the VERDICT intracellular volume fraction can discriminate between Gleason 3 + 3 and 3 + 4 lesions, in contrast to ADC [26].

This study used different DW-MRI techniques to investigate quantitative differences between clinically significant cancer, false positive biopsy results and healthy tissue. The aims were (i) to discriminate false positives from cancer and (ii) to discriminate false positives from normal tissue in an attempt to identify potential diffusion signatures of benign diseases that mimic cancer. We analyzed a total of 38 patients that underwent mp-MRI followed by VERDICT-MRI. We fitted the diffusion models to the VERDICT-MRI data and obtained the ADC from the mp-MRI. The model fitting procedure used deep neural networks (DNNs) for ultra-fast and robust parameter estimation [27]. We compared parameter estimates between both false positives and clinically significant cancer and normal tissue and false positives using statistical tests. We investigated the diagnostic accuracy of different parameters using receiver operating characteristic (ROC) curves, and analyzed the correlation between VERDICT parameters and those from the simpler models. The key contributions of this study are that diffusion MRI models can differentiate between false positives and true cancer, and that models that account for vasculature (IVIM and VERDICT) have the further sensitivity to discriminate false positives from normal tissue. This shows potential for quantitative diffusion MRI to reduce the number of unnecessary invasive biopsies occurring in PCa patients and to identify unique diffusion signatures for a variety of benign pathologies.

## 2. Materials and Methods

### 2.1. Patient Cohort

The study was performed with the approval of the local ethics committee embedded within the INNOVATE clinical trial [25]. The trial is registered with ClinicalTrials.gov, identifier NCT02689271. Thirty-eight men (median age, 67 years; age range, 50–79 years) were recruited, and they provided informed written consent.

The inclusion criteria were as follows:Suspected PCa;Undergoing active surveillance for known PCa.Exclusion criteria included the following:Inability to have an MRI scan, or presence of an artefact that would reduce quality of MRI;Previous hormonal/radiation therapy or surgical treatment for PCa;Biopsy within 6 months prior to the scan.

All patients underwent mp-MRI in line with international guidelines [28] on a 3T scanner, supplemented by VERDICT MRI. The clinical dynamic contrast-enhanced (DCE) part of mp-MRI was performed last, after the VERDICT DW-MRI. After the clinical mp-MRI and VERDICT MRI indicated suspected PCa, all patients underwent targeted transperineal template biopsy of their index lesion, defined as the highest-scoring lesion identified on mp-MRI with Likert scores (3–5) [29]. Specialist genitourinary pathologists evaluated histological specimens stained with hematoxylin and eosin from the biopsy cores. Patients who had a biopsy with diagnoses of atrophy, inflammation, and high-grade PIN (or combinations of these) or clinically significant PCa were retrospectively selected, as shown in Figure 1a. Table 1 presents the clinical and pathological information of the patients.

### 2.2. Image Acquisition

#### 2.2.1. Mp-MRI

All participants underwent mp-MRI with a 3T MRI system (Achieva; Philips, Best, the Netherlands) as part of their standard clinical care. A spasmolytic agent (Buscopan, Boehringer Ingelheim, Ingelheim am Rhein, Germany; 0.2 mg/kg, up to 20 mg) was administered intravenously before imaging to reduce bowel peristalsis. Imaging parameters for the diffusion-weighted echo-planar imaging sequences were as follows: repetition time, 2753 ms; echo time, 80 ms; field of view, 220 × 220 mm; section thickness, 5 mm; no intersection gap; acquisition matrix, 168 × 169; b-values, 0, 150, 500 and 1000 s/mm^2^; and six signals acquired per b-value for signal averaging. The total imaging time for the clinical diffusion-weighted sequences was 5 min and 16 s. ADC maps were calculated by using all b-values except b = 0 to reduce perfusion effects [30], and were calculated with the Camino Diffusion MRI toolkit [31].

#### 2.2.2. VERDICT MRI

VERDICT MRI was performed before dynamic contrast material–enhanced imaging on the same 3T unit as the clinical mp-MRI acquisition. A PGSE sequence was used at five combinations of b-value (in s/mm^2^), gradient duration δ, separation Δ, echo time TE and repetition time TR (in ms) in three orthogonal directions, using a cardiac coil. For each combination, a separate b = 0 image was acquired. Sequences used an echo-planar readout, and imaging parameters were as follows: repetition time, 2482–3945 ms/echo time, 50–90 ms; field of view, 220 × 220 mm; voxel size, 1.3 × 1.3 × 5 mm; no intersection gap; acquisition matrix, 176 × 176; b-values, 90, 500, 1500, 2000 and 3000 s/mm^2^; and six signals acquired per b-value (except for b = 90 s/mm^2^, for which four signals were acquired) for signal averaging. The total imaging time was 12 min and 25 s [32].

### 2.3. Image Analysis

#### 2.3.1. ROIs

Patients were biopsied depending on their mp-MRI score, as reported by two board-certified experienced uroradiologists (reporting more than 2000 prostate MR scans per year). The regions of interest (ROIs) were targeted and drawn by a board-certified study radiologist (S. Singh) using a pictorial report made by the uroradiologist, and confirmed as cancerous or non-cancerous retrospectively by transperineal biopsy. The ROIs were chosen to be as large as possible, while having minimal contamination from surrounding tissue. It was concluded that 19 of the patients had benign pathologies, such as atrophy, inflammation or high-grade PIN, whilst the remaining 19 had cancerous prostate lesions. After a review of the biopsy result confirmed the absence of a tumor on the contralateral side of the peripheral zone for the 19 patients with PCa, ROIs were located for each patient in an area of benign tissue to be used for comparison.

#### 2.3.2. DW-MRI Data Preprocessing

The preprocessing pipeline included denoising of the raw DW-MRI using MP-PCA [33], as implemented within MrTrix3 [34] ‘dwidenoise’, and then correction for Gibbs ringing [35] with custom code in MATLAB (The MathWorks Inc., Natick, Massachusetts, USA). In an effort to reduce possible artefacts caused by patient movement during scanning and eddy current distortions, we applied mutual-information rigid and affine registration using custom code in MATLAB [36].

#### 2.3.3. Mathematical Models

The ADC model is a simple mono-exponential that characterizes the diffusion signal decay as a function of the b-value. It assumes an isotropic Gaussian distribution of water molecule displacements and has one parameter to be estimated: the ADC, *d*. The normalized signal is given by the following:$$S=e^{-bd}$$

The IVIM model is biexponential, with the assumption that the diffusion signal is made up of two non-exchanging compartments of water molecules (one fast and one slow), each following an ADC model. There are three parameters to be estimated: *f*, the volume fraction associated with the fast (‘vascular’) component; *D**, the diffusivity of the ‘fast’ component; and *D*, the diffusivity of the slow (‘cellular’) component [37]. The normalized signal is given by the following:$$S=fe^{-b\left(D+D*\right)}+\left(1-f\right)e^{-bD}$$

The mean signal DKI model relaxes the assumption in the ADC model of Gaussian water dispersion. There are two parameters to be estimated: *D_K_* and *K*. The diffusivity parameter, *D_K_*, is similar to the ADC parameter *d*, whilst the kurtosis parameter, *K*, quantifies the degree of deviation of the dispersion pattern from a Gaussian distribution [38]. The normalized signal is calculated as follows:$$S=e^{-bD_{K}+\frac{1}{6}b^{2}D_{K}^{2}K}$$

The VERDICT model for prostate is the sum of three parametric models, each describing the diffusion magnetic resonance signal in a separate population of water from one of the three components:Signal S_1_ comes from intracellular water trapped within cells (including epithelium);Signal S_2_ comes from extracellular–extravascular water adjacent to but outside cells and blood vessels (including stroma and lumen);Signal S_3_ comes from water in blood undergoing microcirculation in the capillary network.

It is assumed that there is no water exchange between the three tissue compartments. The total signal for the multi-compartment VERDICT model is as follows:$$S=\overset{3}{\underset{i=1}{\sum }}f_{i}S_{i}$$
where *f_i_* is the proportion of signal with no diffusion weighting (b = 0) from water molecules in population *i*, where *i* = *IC*, *VASC* or *EES*; $0\leq f_{i}\leq 1$; and $\overset{3}{\underset{i=1}{\sum }}f_{i}=1$. This results in the following:$$S=f_{VASC}S_{VASC}\left(d_{VASC},b\right)+f_{IC}S_{IC}\left(d_{IC},R,b\right)+f_{EES}S_{EES}\left(d_{EES},b\right)$$
where $f_{VASC}+f_{IC}+f_{EES}=1$.

The VERDICT model used in this work [39] represents the intracellular component as spheres of radius *R* and intra-sphere diffusivity fixed at *d_IC_* = μm^2^/ms; the extracellular-extravascular component as Gaussian isotropic diffusion with effective diffusivity *d_EES_* (Ball); and the vascular component as randomly oriented sticks with intra-stick diffusivity fixed at *d_VASC_* = μm^2^/ms (AstroSticks). In total, there are four model parameters that are estimated by fitting the signal model to DW-MRI data: *f_EES_*, *f_IC_*, *R* and *d_EES_*. The vascular signal fraction, *f_VASC_*, is computed as 1 − *f_IC_* − *f_EES_*, and a cellularity index is computed as *f_IC_/R^3^*. Several previous studies [24,32] have investigated the validity of the assumptions made in this model under the experimental conditions of the optimized DW-MRI acquisition for VERDICT in prostate.

#### 2.3.4. Model Fitting

The IVIM, DKI and VERDICT models were fitted to the DW-MRI data using the signal averaged across three gradient directions. To obtain an ultra-fast and robust parameter estimation, we performed the fitting using a DNN known as a multilayer perceptron (MLP), implemented using the ‘MLPregressor’ in Python scikit-learn 0.23 (https://scikitlearn.org/stable/ (accessed on 12 October 2021)). We chose MLP for this study as it is the simplest deep learning algorithm, and has been used successfully in previous studies for efficient and robust microstructural parameter estimation [40,41]. The input of the DNN is a vector of DW-MRI signals for each combination of b, TE and TR (a total of 10 in this specific case). The DNN consists of three fully connected hidden layers with 150 neurons, each characterized by a linear matrix operation, followed by an element-wise rectified linear unit function (ReLU) and a final regression layer with the number of output neurons equal to the number of tissue parameters to be estimated (i.e., four for the VERDICT model used here). The DNN is optimized by backpropagating the mean squared error (MSE) between ground truth model parameters and DNN predictions. We performed the optimization with the adaptive moment estimation method for 1000 epochs (adaptive learning rate with initial value of 0.001; one update per minibatch of 100 voxels; early stopping to mitigate overfitting; and momentum = 0.9). We normalized the input data to [0, 1] and rescaled the prediction back from the networks.

The DNN was trained using synthetic data, which has been proven to achieve equivalent robustness to real data for deep learning model fitting [42]. We generated 100,000 synthetic DW-MRI signals (split into 80% for training and 20% for validation) using the signal equations above, with different values for the model parameters randomly chosen between biophysically plausible intervals: *f* = [0.01, 0.99], *D* = [0.5, 3] μm^2^/ms and *D** = [0.5, 3] μm^2^/ms for IVIM; *D_K_* = [0.5, 3] μm^2^/ms and *K* = [0.01, 2.99] for DKI; and *f_EES_* = [0.01, 0.99], *f_IC_* = [0.01, 0.99], *R* = [0.01, 15] μm and *d_EES_* = [0.5, 3] μm^2^/ms for VERDICT. We also added Rician noise corresponding to SNR = 35 to consider the effect of experimental noise. For the final parameter computation, we used the DNN at the epoch with minimum validation loss. The creation of the training set and training of the DNN (which was performed only once) took approximately 200 s (1.1 GHz Dual-Core Intel Core M processor). Prediction of the trained DNN for the whole unmasked DW-MRI dataset (roughly 5 × 10^5^ voxels) took approximately 50 s for each subject [27,36,43]. The image analysis pipeline is shown in Figure 1b.

#### 2.3.5. Statistical Analysis

The performance of each parameter for differentiating between the three tissue types was evaluated via a Wilcoxon signed-rank test (preceded by a Shapiro–Wilk test for normality). This was performed using the scipy.stats package [44], and *p* < 0.05 was taken to indicate significance. The data are presented using boxplots, allowing for visualization of the median, interquartile range and any outliers. ROC curves were plotted for all models to compare sensitivity and specificity, and the area under the curve (AUC) was calculated from these to compare the parameters’ utility for tissue type discrimination. We also investigated the correlation between the VERDICT parameters and those from the DKI and IVIM models via the r^2^-value.

## 3. Results

The aim of the first experiment was to analyze differences in parameter estimates in false positive and true positive lesions from the different models. In Figure 2, we see comparisons between parameter estimates from the ADC, IVIM, DKI and VERDICT models in these two tissue types. All models provided discrimination between false positives and true positives: ADC *d* with *p* = 0.002 (Wilcoxon signed-rank test); IVIM *f* with *p* = 0.0002 and *D* with *p* = 0.01; DKI *D_K_* with *p* < 0.0001 and *K* with *p* = 0.0001; and VERDICT *f_IC_* with *p* = 0.001, *f_EES_* with *p* = 0.002 and *d_EES_* with *p* = 0.0004. The *d*, *f*, *D*, *D_K_*, *f_EES_* and *d_EES_* are all lower in cancerous lesions than in false positives, while *K* and *f_IC_* values are higher in cancerous lesions. The positive predictive value (PPV) for discrimination between false positives and true positives is maximized for an ADC *d* of 503; IVIM *f* of 0.1838 and *D* of 0.2795; a DKI *D_K_* of 0.8384 and *K* 0.6446; and VERDICT *f_IC_* of 0.5256, *f_EES_* of 0.1556 and *d_EES_* of 0.1598. High negative predictive values (NPVs) are found for IVIM *f* of 0.4725; DKI *D_K_* of 1.7100; and VERDICT *f_IC_* of 0.0869, *f_EES_* of 0.8510 and *d_EES_* of 3.5629.

The next experiment analyzed differences in parameter estimates between false positive lesions and normal tissue; these are also presented in the boxplots in Figure 2. The only parameters that show statistically significant differences between these tissue types are the VERDICT *f_IC_* (*p* = 0.004), with normal tissue having lower values than false positives, and the IVIM *D* (*p* = 0.02), with higher values in normal tissue than false positives. The PPV for discrimination between false positives and normal tissue is maximized for IVIM *D* of 0.4091; and VERDICT *f_IC_* of 0.3165, *f_EES_* of 0.2072 and *d_EES_* of 1.623. The NPV is maximized for IVIM *D* of 0.8971; DKI *D_K_* of 2.3562 and *K* of 0.3017; and VERDICT *f_IC_* of 0.0877 and *f_EES_* of 0.8940.

Next, we analyzed the parameter maps obtained using the different models, focusing on those parameters with statistically significant differences between true positive and false positive lesions. Figure 3 illustrates parametric maps for two example patients: a 70-year-old with a false positive lesion in the left anterior prostate (atrophy and mild focal chronic inflammation) and a 72-year-old with a Gleason score 4 + 3 tumor in the right posterior peripheral zone (PZ). The parameter maps firstly show clear differences in the data between the true cancer ROIs and the surrounding tissue, improving lesion conspicuity over the mp-MRI images. They also demonstrate that the true positive lesions are more noticeably different from the surrounding healthy tissue than the false positive lesions. We observe that the *f_IC_* and *K* are higher in lesions than the surrounding tissue, whilst for all the other parameters, the values are lower in the lesions than in the surrounding tissue. The VERDICT *f_IC_*, *f_EES_* and *d_EES_* strongly highlight the true positive tumor in comparison to the surrounding benign tissue, showing a clearer difference compared to the other models. The VERDICT *f_IC_* map also significantly highlights the false positive lesion, which is not evident in the other parametric maps.

Following this, we investigated the diagnostic accuracy of the different parameters using ROC curves; we compared the performance of all parameters that can successfully discriminate between true and false positive lesions. Figure 4 presents discrimination between normal tissue and false positives on the left and between false positives and true positives on the right. We observed that the largest AUC value for discrimination between true positives and false positives is found for the DKI *D_K_* (0.9086), followed by IVIM *f* (0.8476), VERDICT *d_EES_* and DKI *K* (0.8338). For the discrimination between false positives and normal tissue, the largest AUC is found for the IVIM *D* (0.7036), followed by the VERDICT *f_IC_* (0.6981).

The final experiment investigated the correlations between the VERDICT *f_IC_*, *R*, *f_EES_* and *f_VASC_*; IVIM *D*, *f* and *D**; and DKI *D_K_* and *K* for each voxel within the ROIs of all patients; the strongest correlations are presented in Figure 5. The color coding distinguishes individual patients, with the cancer ROIs shown as circles, whilst the benign ROIs are shown as crosses. We observed similar trends for *D_K_* and *D*: negative correlation with *f_IC_* and positive correlation with *f_EES_* and *d_EES_*. *K* showed a strong positive correlation with *f_IC_* and negative correlation with *f_EES_*. Finally, *D** showed a negative correlation with *f_EES_* and a positive correlation with *f_VASC_*. 

## 4. Discussion

Benign prostate pathologies, such as atrophy, inflammation and high-grade PIN, can cause false positives on mp-MRI by having signal characteristics that mimic PCa. We aimed to firstly differentiate false positives from true cases of PCa, and then to discriminate false positives from normal tissue, using various diffusion MRI models. We analyzed the clinical ADC and then fitted the IVIM, DKI and VERDICT models to the acquired DW-MRI from 38 patients using a deep learning approach. We then compared the model parameter estimates between tissue types using statistical tests, to draw conclusions about the diagnostic utility of the different models for characterizing false positive cases of PCa.

Our results showed that all models are able to discriminate false positives from true positives. The strongest statistical significance is observed for the DKI *D_K_* (*p* < 0.0001) and *K* (*p* = 0.0001), followed by the IVIM *f* (*p* = 0.0002); VERDICT *d_EES_* (*p* = 0.0004), *f_IC_* (*p* = 0.001) and *f_EES_* (*p* = 0.002); ADC *d* (*p* = 0.002); and IVIM *D* (*p* = 0.01). The *D_K_*, *d_EES_*, *D*, *f_EES_*, *f* and *d* are lower in true cancer than in false positives, whilst *K* and *f_IC_* are higher. This reflects the reduced diffusivity and larger deviations from Gaussian dispersion due to the increased cellularity in prostate carcinoma than in non-cancer diseases [45]. No significant differences were found in IVIM *D*,* VERDICT *f_VASC_*, *R* or Cellularity. The best diagnostic performance was found for *D_K_* (AUC = 0.9086), followed by *f* (AUC = 0.8476), *d_EES_* and *K* (AUC = 0.8338). We also observed that the VERDICT maps emphasize the true positive lesion in comparison to the surrounding tissue most clearly. Our finding of significantly decreased ADC values in true positives agrees with work by Stavrinides et al., who concluded that ADC could predict clinically significant PCa in biopsy-naive men with indeterminate lesions [46]. Falaschi et al. similarly found a lower ADC ratio in tumors than false positives; however, they could not draw any firm conclusions about the usefulness of the ADC ratio in detecting cancer [47].

The VERDICT *f_IC_* and IVIM *D* are the only parameters that are able to discriminate false positives from normal tissue. Significantly higher values of *f_IC_* (*p* = 0.004) and significantly lower values of *D* (*p* = 0.02) were found in false positives, thus agreeing with histological findings of increased cellularity and reduced diffusivity in the false positive disease types in comparison to healthy tissue [48]. The diagnostic accuracy is highest for *D* (AUC = 0.7036), closely followed by *f_IC_* (AUC = 0.6981). We did not find any significant differences in ADC or DKI parameters, potentially due to the models’ limited biological specificity [45]. Both the IVIM and VERDICT models account for vasculature, which may explain their increased diagnostic efficacy for this discrimination, in agreement with Wang et al. [49]. This indicates potential for these models to identify unique diffusion signatures of diseases that mimic PCa.

This study demonstrated the utility of various diffusion MRI models for tackling the specific diagnostic obstacle of false positive cases of PCa. All diffusion models (ADC, IVIM, DKI and VERDICT) were able to discriminate false positives from true positives; however, only VERDICT and IVIM revealed significant differences between false positives and normal tissue. The VERDICT parameters also allow for inferences to be made about microstructural differences between tissue types, as shown by the correlation analysis. We observed negative a correlation of *D_K_* and *D* with *f_IC_*, but a positive correlation with *f_EES_*. This is expected, as diffusivities tend to decrease as the proportion of water trapped in cells increases. The DKI *K* showed a strong positive correlation with *f_IC_* and negative correlation with *f_EES_*, which is also expected due to larger deviations from Gaussian dispersion as the proportion of water trapped in cells decreases. These observations emphasize VERDICT’s enhanced biological specificity, a finding which is supported by the *f_IC_* discriminating between all tissue types with stronger statistical significance than *D*.

The main limitation of this work was the number of participants—a larger patient cohort would allow us to improve the significance level of the results obtained, as well as potentially enable the identification of unique diffusion signatures for the different benign pathologies. However, we still achieved statistical significance for all the diffusion models considered. Moreover, this analysis was performed on retrospective data with an acquisition protocol optimized for VERDICT, and this may mean that the choice of b-values was not optimal for IVIM and DKI parameter estimation. In addition, the range of benign pathologies in our study was limited to atrophy, inflammation and high-grade PIN, but the inclusion of others, such as benign prostatic hyperplasia, may allow for more comprehensive benign disease characterization. Future work will increase the size of the patient cohort, encompassing a wider range of prostatic diseases and potentially allowing for the identification of transitionary periods in which benign diseases become malignant. We will also investigate more sophisticated models that take relaxation effects into account, such as relaxed-VERDICT [50].

## 5. Conclusions

In this work, we demonstrated that quantitative diffusion MRI (ADC, IVIM, DKI and VERDICT) can successfully discriminate false positive prostate lesions from cancerous tumors, showing the potential to avoid unnecessary biopsies. The best diagnostic accuracy for discriminating false positives and true positives was observed for the DKI *D_K_*. Among the different diffusion models, only VERDICT and IVIM were able to also differentiate false positive lesions from normal prostate tissue, correctly identifying benign diseases that mimic cancer. This work was primarily limited by the small size of the patient cohort; future work will include more patients with a wider range of benign pathologies.

## Figures and Tables

**Figure 1 diagnostics-12-01631-f001:**
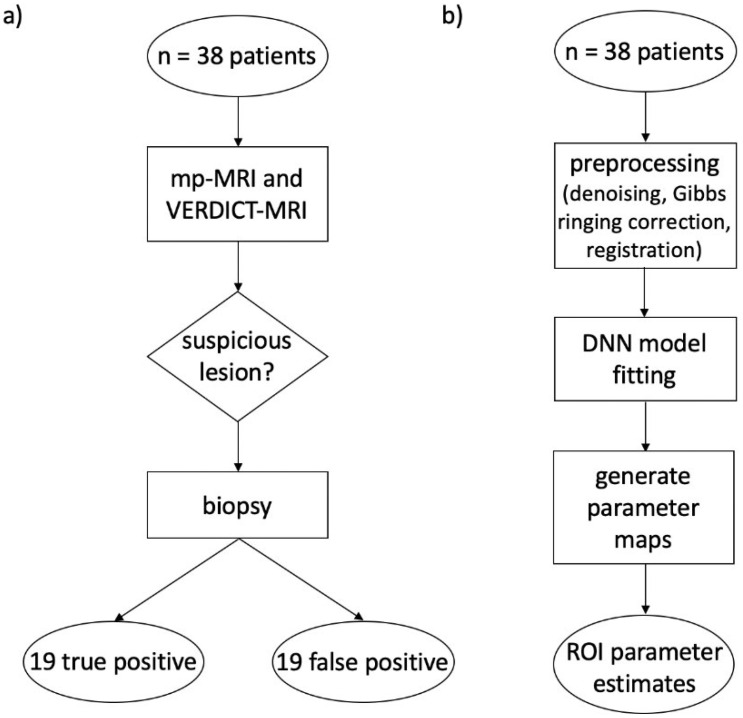
Flowcharts describing full methodology of study. (**a**) Image acquisition pipeline to determine nature of patient lesion. (**b**) Image analysis pipeline for deep learning parameter estimation used to fit IVIM, DKI and VERDICT models.

**Figure 2 diagnostics-12-01631-f002:**
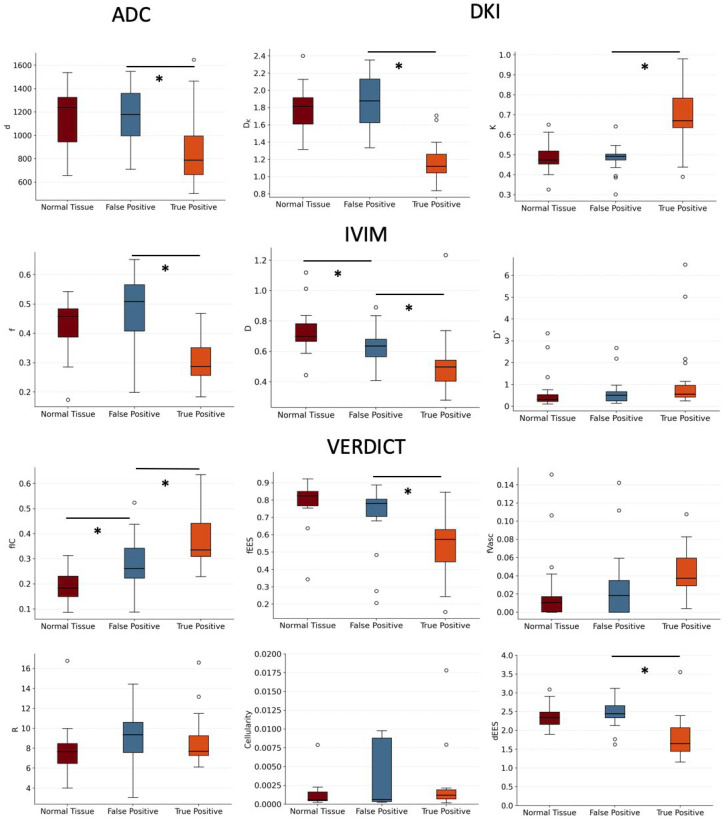
Boxplots showing the parameter estimates obtained using the ADC, DKI, IVIM and VERDICT models. We observed significant differences between true and false positives in the ADC *d* (*p* = 0.002); IVIM *D* (*p* = 0.01) and *f* (*p* = 0.0002); DKI *D_K_* (*p* < 0.0001) and *K* (*p* = 0.0001); and VERDICT *f_IC_* (*p* = 0.001), *f_EES_* (*p* = 0.002) and *d_EES_* (*p* = 0.0004). The *d*, *D_K_*, *D*, *f*, *f_EES_* and *d_EES_* are all lower in true positives than false positives, while *K* and *f_IC_* are higher. We also found statistically significant differences between false positives and normal tissue for the VERDICT *f_IC_* (*p* = 0.004), with higher values in false positive lesions than in normal tissue, and the IVIM *D* (*p* = 0.02) with lower values in false positive lesions than in normal tissue. Outliers are denoted by a circle and asterisks indicate statistical significance.

**Figure 3 diagnostics-12-01631-f003:**
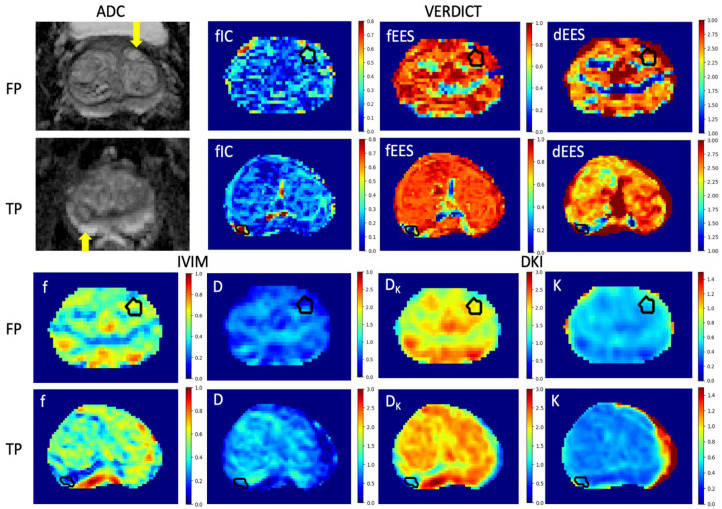
Parametric maps obtained using the IVIM, DKI and VERDICT models in a 70-year-old patient with a false positive lesion and a 72-year-old patient with a true positive lesion. Only the parameters which successfully differentiated between the two lesion types are included, as well as the clinical ADC maps. We observed that the VERDICT maps highlight the true positive lesion most conspicuously, and the VERDICT *f_IC_* also distinguishes the false positive lesion from the surrounding tissue.

**Figure 4 diagnostics-12-01631-f004:**
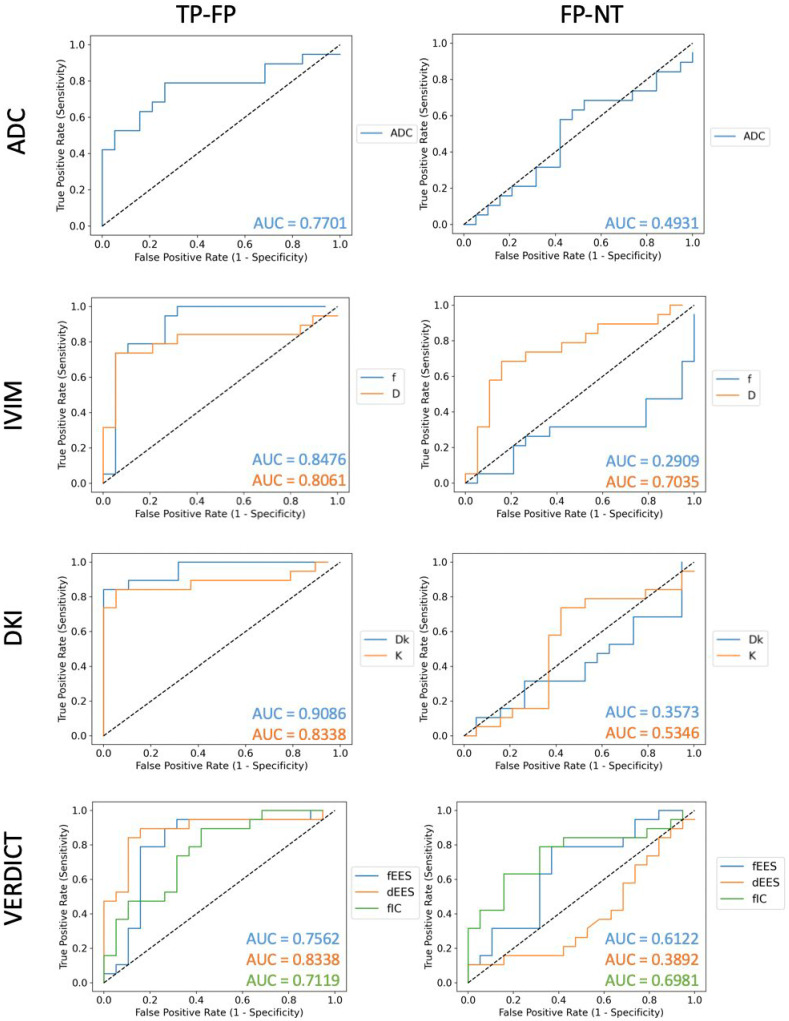
ROC curves for ADC, DKI, IVIM and VERDICT parameters—those on the left are for discriminating true (TP) and false positives (FP), and those on the right are for discriminating false positives and normal tissue (NT). We observed that the largest AUC for discrimination between true and false positives is achieved by the DKI *D_K_* (AUC = 0.9086). The largest AUC for discrimination between false positives and normal tissue is achieved by the IVIM *D* (AUC = 0.7036), closely followed by the VERDICT *f_IC_* (AUC = 0.6981).

**Figure 5 diagnostics-12-01631-f005:**
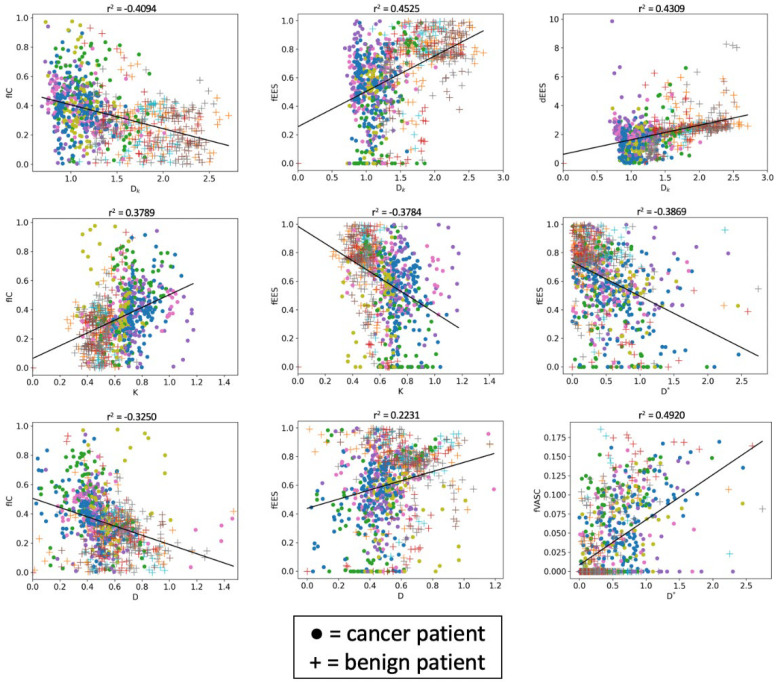
Scatter plots showing the correlation between VERDICT parameters and the DKI and IVIM parameters. For *D_K_* and *D*, we observed a negative correlation with *f_IC_* and positive correlation with *f_EES_* and *d_EES_*. *K* shows a positive correlation with *f_IC_* and negative correlation with *f_EES_*. Finally, *D** shows a negative correlation with *f_EES_* and positive correlation with *f_VASC_*.

**Table 1 diagnostics-12-01631-t001:** Clinical and pathological information of 38 patients included in the study (19 with no/clinically insignificant cancer and 19 with clinically significant cancer). The median age and PSA/PSAD results for each cohort are presented, along with the biopsy results/Gleason scores. The false positive patient cohort had combinations of the three disease types considered.

	No/Clinically Insignificant Cancer	Clinically Significant Cancer
Age (Median)	65	66
PSA (Median)	6.91	14.22
PSAD (Median)	0.113	0.426
Biopsy Result	Atrophy: 16	3 + 3: 1
	Inflammation: 13	3 + 4: 7
	High-grade PIN: 5	4 + 3: 9
		4 + 4: 1
		4 + 5: 1

## Data Availability

Data are available upon request.

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
