# Peer review of "Differentiating False Positive Lesions from Clinically Significant Cancer and Normal Prostate Tissue Using VERDICT MRI and Other Diffusion Models"

_diagnostics, 2022, doi:10.3390/diagnostics12071631_

Round 1
Reviewer 1 Report
The authors presented an interesting study to differentiate false positive lesions from significant cancer and normal tissue for prostate cancer patients using three different MRI diffusion models. However, I have the following concerns.
1. The authors could elaborate more details on the model fitting part. Why did the authors choose this DNN for model fitting? For time-saving purposes, the training of the DNN could take time. And the authors used synthetic DW-MRI signals, how did the authors make sure the model could provide robust parameter estimation? Also, more details could be provided for the synthetic DW-MRI signals (how were they created, any example data, etc).
2. The sample size was too small. The ROC analysis was based on this small dataset, instead of an independent testing set. In this case, I am not sure how useful these high AUCs were. If a threshold could be determined based on each useful parameter/or the combination of these useful parameters, and an independent validation can be performed using a separate dataset, the results can be more convincing.
Author Response
We would like to thank the Editor for the opportunity to revise our manuscript. Please see below, our point-by-point response (in red) to each of the Reviewers’ concerns (in black).
Reviewer 1:
Comments to the author:
The authors presented an interesting study to differentiate false positive lesions from significant cancer and normal tissue for prostate cancer patients using three different MRI diffusion models. However, I have the following concerns.
Point 1: The authors could elaborate more details on the model fitting part. Why did the authors choose this DNN for model fitting? For time-saving purposes, the training of the DNN could take time. And the authors used synthetic DW-MRI signals, how did the authors make sure the model could provide robust parameter estimation? Also, more details could be provided for the synthetic DW-MRI signals (how were they created, any example data, etc).
Response 1:
We added information about the choice of DNN to page 6, paragraph 1, including references to doi: 10.1109/ISBI.2019.8759161 and doi: 10.1109/TMI.2016.2551324.
“We chose MLP for this study as it is the simplest deep learning algorithm and has been used successfully in previous studies for efficient and robust microstructural parameter estimation [40,41].”
We added information on time for DNN training and prediction to page 6, paragraph 2.
“The creation of the training set and training of the DNN (which was only performed once) took approximately 200 seconds (1.1 GHz Dual-Core Intel Core M processor). Prediction of the trained DNN for the whole unmasked DW-MRI dataset (roughly 5 × 105 voxels) took approximately 50 seconds for each subject [27,36,43].”
We added information on the robustness of parameter estimation using synthetic training data to page 6, paragraph 2, including reference to doi: 10.1101/2020.10.20.347625.
“The DNN is trained using synthetic data, which has been proven to achieve equivalent robustness to real data for deep learning model fitting [42].”
Information on creation of synthetic DW-MRI signals: included on page 6, paragraph 2.
“We generate 100,000 synthetic DW-MRI signals (split into 80% for training and 20% for validation) using the signal equations above with different values for the model parameters randomly chosen between biophysically plausible intervals: f = [0.01, 0.99], D = [0.5, 3] mm2/ms and D* = [0.5, 3] mm2/ms for IVIM, DK = [0.5, 3] mm2/ms and K = [0.01, 2.99] for DKI and fEES = [0.01, 0.99], fIC = [0.01, 0.99], R = [0.01, 15] mm and dEES = [0.5, 3] mm2/ms for VERDICT. We also add Rician noise corresponding to SNR=35 to consider the effect of experimental noise."
Added that the synthetic signals are split into 80% for training and 20% for validation.
Point 2: The sample size was too small. The ROC analysis was based on this small dataset, instead of an independent testing set. In this case, I am not sure how useful these high AUCs were. If a threshold could be determined based on each useful parameter/or the combination of these useful parameters, and an independent validation can be performed using a separate dataset, the results can be more convincing.
Response 2: To address this comment on sample size, we have added 15 more patients to the dataset (7 false positive/benign and 8 true positive), resulting in a sample size of 38 overall, instead of 23 patients analysed previously. We note that our previous results still stand, with improved statistical significance. Also, we have added positive/negative predictive value analysis to the study as suggested by the reviewer (paragraphs 1 and 2, page 7).
Reviewer 2 Report
The authors presented a method to discriminate against false-positive cancer cases. I have the following comments:
1) In the current form Abstract is missing existing methods information, how the existing methods (without the proposed methods ) are actually lacking for diagnostic purposes?
2) The Abstract should end with achieving the proposed method with numerical performance eleboration.
3) I would suggest adding more references related to cancer diagnosis in the introduction
4) A clear flow chart of the complete proposed method is essentially required to delve deep into the proposed method
5) Mention the key contribution points of the paper at the end of the introduction
6) mention the Hardware/ software used for training and testing the model including the hyperparameters.
7) The conclusion is missing with limitations of the proposed method and future directions
Author Response
We would like to thank the Editor for the opportunity to revise our manuscript. Please see below, our point-by-point response (in red) to each of the Reviewers’ concerns (in black).
Comments to the author:
The authors presented a method to discriminate against false-positive cancer cases. I have the following comments:
Point 1: In the current form Abstract is missing existing methods information, how the existing methods (without the proposed methods) are actually lacking for diagnostic purposes?
Response 1: We have made the relevant changes at the beginning of the abstract, page 1.
“False positives on multiparametric (mp)-MRI result in a large number of unnecessary invasive biopsies in men with clinically insignificant diseases. This study investigates whether quantitative diffusion MRI can differentiate between false positives, true positives and normal tissue non-invasively.”
We have also included this information in the introduction, page 2, paragraph 1.
“This technique has high sensitivity (90%) but moderate specificity (50%), translating to a high rate of false positive cases [3]. This results in 1 in 2 men undergoing mp-MRI having unnecessary uncomfortable biopsies and risking the associated side effects for benign conditions or clinically insignificant cancer [4].”
Point 2: The Abstract should end with achieving the proposed method with numerical performance elaboration.
Response 2: We have added the p-values of best performing parameters to the abstract, page 1.
“Significant differences (p < 0.05) between true positive and false positive lesions were found in ADC, IVIM perfusion fraction (f) and diffusivity (D), DKI diffusivity (DK) (p = 0.0000) and kurtosis (K) and VERDICT intracellular volume fraction (fIC), extracellular-extravascular volume fraction (fEES) and diffusivity (dEES) values. Significant differences between false positives and normal tissue were found for the VERDICT fIC (p = 0.004) and IVIM D.”
Point 3: I would suggest adding more references related to cancer diagnosis in the introduction
Response 3: We have added doi: 10.1016/j.ajur.2018.11.007 to the start of the introduction, page 1.
“Prostate cancer (PCa) is traditionally diagnosed via digital rectal inspection (DRE) and prostate-specific antigen (PSA) test, followed by transrectal ultrasound (TRUS)-guided biopsy [1].”
Point 4: A clear flow chart of the complete proposed method is essentially required to delve deep into the proposed method.
Response 4: A flow chart of the complete method has been created and presented as Figure 1 in the manuscript, page 6.
Point 5: Mention the key contribution points of the paper at the end of the introduction
Response 5: We have added the key contribution points at the end of the introduction, page 3.
“The key contributions of this study are that diffusion MRI models can differentiate between false positives and true cancer, and that models that account for vasculature (IVIM, VERDICT) have the further sensitivity to discriminate false positives from normal tissue. This shows potential for quantitative diffusion MRI to reduce the number of unnecessary invasive biopsies occurring in PCa patients and to identify unique diffusion signatures for a variety of benign pathologies.”
Point 6: Mention the Hardware/ software used for training and testing the model including the hyperparameters.
Response 6: We have added information about the hardware to page 6, paragraph 2.
“1.1 GHz Dual-Core Intel Core M processor”.
We have included information on the software on page 6, paragraph 1 (reference to Python scikit-learn 0.23).
We have added details of the hyperparameters to page 6, paragraph 1.
“We perform the optimisation with the adaptive moment estimation method for 1000 epochs (adaptive learning rate with initial value of 0.001; one update per mini-batch of 100 voxels; early stopping to mitigate overfitting; and momentum = 0.9). We normalized the input data to [0,1] and re-scaled the prediction back from the networks.”
Point 7: The conclusion is missing with limitations of the proposed method and future directions
Response 7: We have added limitations and future work at the end of the conclusions, page 11.
“This work is primarily limited by the small size of the patient cohort - future work will include more patients with a wider range of benign pathologies.”
Reviewer 3 Report
Authors aimed to investigate whether quantitative diffusion MRI can improve differentiation between false positives, true positives and normal tissue.
1. In the abstract, how many patients belong to “e (2) atrophy/inflammation/high-grade prostatic intraepithelial neoplasia (PIN) - false positive.”?
2. Furthermore, please, describe their natural courses, e.g. follow-up interval or the timing of malignant transformation since initial diagnosis.
3. Can authors establish cutoff points to achieve high positive predictive value or negative predictive value?
4. Please, discuss the possibility to develop “intermediate zone”, which is more useful from the clinical viewpoint.
Author Response
We would like to thank the Editor for the opportunity to revise our manuscript. Please see below, our point-by-point response (in red) to each of the Reviewers’ concerns (in black).
Reviewer 3:
Comments to the author:
Authors aimed to investigate whether quantitative diffusion MRI can improve differentiation between false positives, true positives and normal tissue.
Point 1: In the abstract, how many patients belong to “e (2) atrophy/inflammation/high-grade prostatic intraepithelial neoplasia (PIN) - false positive.”?
Response 1: We have added this information to the abstract, page 1.
“The patients were categorised into two groups following biopsy: (1) significant cancer - true positive, 19 patients (2) atrophy/inflammation/high-grade prostatic intraepithelial neoplasia (PIN) - false positive, 19 patients.”
Point 2: Furthermore, please, describe their natural courses, e.g. follow-up interval or the timing of malignant transformation since initial diagnosis.
Response 2: We have added this information to the introduction with reference to doi: 10.1038/modpathol.3800053, page 2, paragraph 2.
“High-grade PIN represents the pre-invasive end of the range of cellular proliferations within the lining of prostatic ducts and acini and is considered the most likely precursor of PCa, with most patients developing carcinoma within 10 years [8].”
Point 3: Can authors establish cutoff points to achieve high positive predictive value or negative predictive value?
Response 3: We have conducted positive/negative predictive value analysis and added this information to the results section, paragraphs 1 and 2, page 7.
“The positive predictive value (PPV) for discrimination between false positives and true positives is maximized for an ADC d value of 503, IVIM f value of 0.1838 and D value of 0.2795, a DKI DK value of 0.8384 and K value 0.6446 and VERDICT fIC value of 0.5256, fEES value of 0.1556 and dEES value of 0.1598. High negative predictive values (NPV) are found for IVIM f threshold of 0.4725, DKI DK value of 1.7100 and VERDICT fIC of 0.0869, fEES value of 0.8510 and dEES value of 3.5629.”
“The PPV for discrimination between false positives and normal tissue is maximized for IVIM D value of 0.4091 and VERDICT fIC value of 0.3165, fEES value of 0.2072 and dEES value of 1.623. The NPV is maximized for IVIM D value of 0.8971, DKI DK value of 2.3562 and K value of 0.3017 and VERDICT fIC value of 0.0877 and fEES value of 0.8940.”
Point 4: Please, discuss the possibility to develop “intermediate zone”, which is more useful from the clinical viewpoint.
Response 4: We thank the reviewer for this interesting point. In order to identify an “intermediate zone”, which could be defined as a transitionary period during which these benign diseases transform to malignant, we would require a significant amount of data from specific subcategories of the benign diseases and follow ups, that is beyond the scope of this study. However we will include it in our future work, which we have added this to the end of the discussion, page 12 paragraph 4.
“Future work will increase the size of the patient cohort, encompassing a wider range of prostatic diseases and potentially allowing for the identification of transitionary periods in which benign diseases transform to malignant.”
Round 2
Reviewer 1 Report
The authors made great efforts to address my questions and improve the manuscript. I only have some minor comments:
1. It's better to write p=0.0000 as p<<0.0001
2. If the ROC analysis could be performed on a separate dataset (independent dataset), the results would be more convincing.
Author Response
We would like to thank the Editor for the opportunity to revise our manuscript, and all the reviewers. Please see below, our point-by-point response (in red) the Reviewers’ concerns (in black).
Reviewer 1:
Comments to the author:
The authors made great efforts to address my questions and improve the manuscript. I only have some minor comments:
Point 1: It's better to write p=0.0000 as p<<0.0001
Response 1: This has been changed in the abstract (page 1, paragraph 1), the results (page 7, paragraph 2) and the discussion (page 11, paragraph 2).
Point 2: If the ROC analysis could be performed on a separate dataset (independent dataset), the results would be more convincing.
Response 2: We were able to identify an additional 15 patients and increase our final cohort. We decided not to treat this as a separate dataset as it is not a large enough cohort size. We would also like to clarify that our dataset is an independent test dataset, and the DNN was only used for the fitting and not for identifying and characterising lesions. We trained the DNN for model fitting on synthetic data, so all patients included in the study are part of the independent ‘test’ dataset.
Reviewer 2 Report
The authors adequately responded to the comments. I vote for acceptance of this paper in current form.
Author Response
We would like to thank the Reviewer for their constructive feedback.
Reviewer 3 Report
Authors addressed raised points appropriately.
Author Response

(The authors gave the same response as above.)
